# Metropolis-CVAE: Bootstrapping Labels for Bayesian Inference via Semi-Supervised Conditional Variational Autoencoders

## Abstract

In Bayesian parameter estimation, models make simplifying assumptions to make parameter inference feasible. If learned inference methods are trained using data simulated by models, however, distributional differences between simulated and observed data may lead to biased inference results on the observed data. In this work, we introduce a semi-supervised learned Bayesian inference method which makes use of both simulated data – for which the underlying parameters are known by construction – and unlabeled data, which may depend on nuissance parameters not captured by the simulation procedure. A conditional variational autoencoder (CVAE) is trained to perform approximate inference simultaneously on the sets of labeled simulated data and unlabeled data, where the unlabeled data is initialized with arbitrary *pseudo* labels. At each training iteration, new candidate pseudo labels are drawn from the CVAE posterior and the pseudo labels are updated using the Metropolis-Hastings algorithm. This process results in a Markov chain of bootstrapped pseudo labels for each unlabeled datum, effectively performing online Markov chain Monte Carlo (MCMC) inference wherein the proposal distribution is a CVAE informed by labeled simulated data, producing proposals which are increasingly likely to be accepted as training proceeds. The resulting CVAE is able to efficiently produce samples from the posterior distributions of both the simulated and unlabeled data, implicitly marginalizing over nuissance parameters in the unlabeled data. We demonstrate the effectiveness of this method in magnetic resonance imaging (MRI) where MCMC is computationally impractical to due the (3+1)D nature of the images, showing improvement against traditional MCMC inference in both speed and posterior quality.

## 1   Introduction

Bayesian parameter estimation methods are robust techniques for quantifying properties of a system that cannot be observed directly [1]. In order to estimate such parameters, one first needs to develop a model of the phenomena to be studied. This process requires deep domain-specific knowledge. For all but the most basic of systems, a series of simplifying assumptions on the system are required to make parameter inference tractable. Typically, tractable is synonymous with not being unreasonably expensive to compute. Examples of such methodologies include perturbation theory, in which models containing Taylor series expansions drop higher order terms; mean-field theory, in which interactions involving many degrees of freedom are replaced by averaged approximations; and (stochastic) differential equations, when used as continuous limits of discrete stochastic processes. The price one pays for utilizing a given approximation is highly problem dependent. When approximate models are used for parameter inference, the mismatch between model predictions and data may propagate

through the inference process and lead to bias and other misestimations of the inferred parameters. In this work, we are interested in the following research question: can a deep learning model be trained to perform inference, while incorporating both labeled data generated from a well-understood model, and unlabeled data which contains additional complex structure which cannot be modeled?

Here, we present Metropolis conditional variational autoencoders (Metropolis-CVAEs). Metropolis-CVAEs combine traditional CVAEs with the Metropolis-Hastings Markov chain monte carlo (MCMC) inference method. The resulting networks combine the ability of CVAEs to learn to perform rapid approximate Bayesian inference from labeled data with traditional MCMC methodology which requires only a likelihood model and a prior distribution over the parameters of interest. The Metropolis-CVAE initializes the unlabeled data with pseudo labels drawn from the prior distribution and, informed by the labeled data, iteratively improves the pseudo labels throughout training. We demonstrate the effectiveness of the Metropolis-CVAE network compared to traditional MCMC methods on an inference problem from magnetic resonance imaging (MRI) which is inherently computationally challenging due to the (3+1)D nature of the data.

## 2 Methods

### 2.1 Related work

In the pioneering work by Sohn et al. [2], conditional variational autoencoders were introduced by considering a variational lower bound to the conditional log-likelihood $\log p(\mathbf{y}|\mathbf{x})$ of labels $\mathbf{y}$ given corresponding data $\mathbf{x}$. Ideally, to perform data-driven approximate inference, one would like to train a network to learn to maximize the conditional log-likelihood directly. However, this is known to be an intractable problem. To mitigate this issue, the stochastic gradient variational Bayes (SGVB) framework is employed. In SGVB, a latent space which factorizes the marginal likelihood is introduced via $p(\mathbf{y}|\mathbf{x}) = \int_{\mathbf{z}} d\mathbf{z}\, p(\mathbf{z}|\mathbf{x})p(\mathbf{y}|\mathbf{z}, \mathbf{x})$, as well as a recognition distribution $q(\mathbf{z}|\mathbf{x}, \mathbf{y})$. The conditional log-likelihood is then maximized indirectly via maximizing the variational lower bound

$$\log p(\mathbf{y}|\mathbf{x}) \geq -\mathrm{KL}\left(q(\mathbf{z}|\mathbf{x}, \mathbf{y}) \,\|\, p(\mathbf{z}|\mathbf{x})\right) + \mathbb{E}_{q(\mathbf{z}|\mathbf{x}, \mathbf{y})}\left[\log p(\mathbf{y}|\mathbf{x}, \mathbf{z})\right]$$
$$:= -\mathcal{L}(\mathbf{x}, \mathbf{y}) \tag{1}$$

where KL is the Kullback-Leibler divergence. The interpretation as a conditional autoencoder is as follows: let $p(\mathbf{z}|\mathbf{x})$, $q(\mathbf{z}|\mathbf{x}, \mathbf{y})$, and $p(\mathbf{y}|\mathbf{x}, \mathbf{z})$ be parameterized by deep neural networks $E_1(\mathbf{z}|\mathbf{x})$, $E_2(\mathbf{z}|\mathbf{x}, \mathbf{y})$, and $D(\mathbf{y}|\mathbf{x}, \mathbf{z})$, respectively. Then, $E_1$ and $E_2$ can be viewed as encoders which map their inputs into distributions over the latent variables $\mathbf{z}$. $D$ can be viewed as a decoder which maps stochastic latent representations $\mathbf{z}$ and data $\mathbf{x}$ into posterior distributions over the labels $\mathbf{y}$. Hence, $\mathbf{y}$ is conditionally autoencoded via the encoder-decoder pipeline $\mathbf{z} \sim E_2(\mathbf{z}|\mathbf{x}, \mathbf{y}) \rightarrow \mathbf{y} \sim D(\mathbf{y}|\mathbf{x}, \mathbf{z})$. At inference time, posterior samples are similarly drawn via $\mathbf{z} \sim E_1(\mathbf{z}|\mathbf{x}) \rightarrow \mathbf{y} \sim D(\mathbf{y}|\mathbf{x}, \mathbf{z})$. The CVAE – the triplet of networks $(E_1, E_2, D)$ – is trained by minimizing $\mathcal{L}$ over the CVAE parameters, thereby maximizing the variational lower bound on $\log p(\mathbf{y}|\mathbf{x})$.

The CVAE approach by Sohn et al. [2] is designed for supervised learning problems. Earlier work of a similar vein by Kingma et al. [3] introduces a semi-supervised framework in which labeled data $(\mathbf{x}_\ell, \mathbf{y})$ is used to infer labels for unlabeled data $\mathbf{x}_u$ via the minimization of a two-term loss function. The first term of the loss function is a supervised loss over $(\mathbf{x}_\ell, \mathbf{y})$ samples, similar to Equation 1, derived from a variational lower bound on the joint log-likelihood $\log p(\mathbf{x}_\ell, \mathbf{y})$. The second term is an unsupervised loss over $\mathbf{x}_u$ samples which treats label inference as a data imputation task. Specifically, the unknown label is treated as a parameter over which posterior inference is performed; the resulting loss is a variational lower bound on the log-likelihood $\log p(\mathbf{x}_u)$.

While the semi-supervised method of Kingma et al. is an elegant approach to discovering labels, it is not quite suitable for inferring labels for out of distribution data. This is due to the implicit assumption that relationships between $\mathbf{x}_\ell$ and $\mathbf{y}$ learned from the joint distribution $p(\mathbf{x}_\ell, \mathbf{y})$ generalize to data from $p(\mathbf{x}_u)$. In fact, this assumption is made explicit via an extended objective function which adds a regularization term $\mathbb{E}_{\widetilde{p}_\ell(\mathbf{x}, \mathbf{y})}\left[-\log q(\mathbf{y}|\mathbf{x}_\ell)\right]$ over the empirical distribution $\widetilde{p}_\ell(\mathbf{x}, \mathbf{y})$ of labeled data. This penalty encourages the learned posterior distribution $q(\mathbf{y}|\mathbf{x})$ to generate labels for $\mathbf{x}_u$ by extrapolating from the relationships it discovers between $(\mathbf{x}_\ell, \mathbf{y})$ pairs. This will naturally lead to biased labels for $\mathbf{x}_u$ when the distribution underlying $\mathbf{x}_u$ differs from that of $\mathbf{x}_\ell$.

Gabbard et al., who made use of CVAEs to accelerate inference for an application in gravitational wave astronomy [4], presented an alternate view of Equation 1. Gabbard et al. begin by aiming to minimize the expected cross-entropy

$$\mathcal{H} := -\,\mathbb{E}_{p(\mathbf{x})}\left[\int_{\mathbf{y}} \mathrm{d}\mathbf{y}\, p(\mathbf{y}|\mathbf{x})\log \hat{p}(\mathbf{y}|\mathbf{x})\right] \tag{2}$$

over the data distribution $p(\mathbf{x})$ between the true posterior $p(\mathbf{y}|\mathbf{x})$ and approximate posterior $\hat{p}(\mathbf{y}|\mathbf{x})$. Employing the SGVB framework and letting $\hat{p}(\mathbf{y}|\mathbf{x}) = \int_{\mathbf{z}} \mathrm{d}\mathbf{z}\, \hat{p}(\mathbf{z}|\mathbf{x})\hat{p}(\mathbf{y}|\mathbf{z}, \mathbf{x})$, it follows from 1 that

$$\mathcal{H} \leq \mathbb{E}_{p(\mathbf{x})}\left[\int_{\mathbf{y}} \mathrm{d}\mathbf{y}\, p(\mathbf{y}|\mathbf{x})\, \mathcal{L}(\mathbf{x}, \mathbf{y})\right]. \tag{3}$$

Applying Bayes' theorem, we equivalently have that

$$\mathcal{H} \leq \mathbb{E}_{p(\mathbf{x})}\, \mathbb{E}_{p(\mathbf{y}|\mathbf{x})}\, [\mathcal{L}(\mathbf{x}, \mathbf{y})] \tag{4}$$
$$= \mathbb{E}_{p(\mathbf{y})}\, \mathbb{E}_{p(\mathbf{x}|\mathbf{y})}\, [\mathcal{L}(\mathbf{x}, \mathbf{y})] \tag{5}$$
$$= \mathbb{E}_{p(\mathbf{x},\mathbf{y})}\, [\mathcal{L}(\mathbf{x}, \mathbf{y})]. \tag{6}$$

Therefore, maximizing the variational lower bound to $\log p(\mathbf{y}|\mathbf{x})$ over a dataset of $(\mathbf{x}, \mathbf{y})$ pairs, as in Equation 1, is equivalent to minimizing the expected cross-entropy via Equation 6. The interpretation of Equations 4 and 5, however, will prove useful for remedying the issue of label inference for out of distribution data.

## 2.2 Theoretical contributions

Equation 5 is a natural framework for using CVAEs to perform inference on simulated data with known labels [4]. Suppose labels $\mathbf{y} \sim p(\mathbf{y})$ are sampled from a prior distribution and $\mathbf{x} \sim p(\mathbf{x}|\mathbf{y})$ is subsequently given by a (possibly stochastic) model function $\mathbf{x} = f(\mathbf{y})$. Then, Equation 5 corresponds to minimizing the average CVAE loss $\mathcal{L}(\mathbf{x}, \mathbf{y})$ over pairs of simulated data $(\mathbf{x} = f(\mathbf{y}), \mathbf{y})$.

The novel contribution of this work stems from of Equation 4. Using this formulation directly would require sampling from the posterior $p(\mathbf{y}|\mathbf{x})$, which is our stated objective. However, this can be circumvented by making the observation that if we could construct a Markov chain of *pseudo* labels $\widetilde{\mathbf{y}}$ during training, such that the stationary distribution of the sequence $(\widetilde{\mathbf{y}}_k)_{k\in\mathbb{N}}$ was $p(\mathbf{y}|\mathbf{x})$, then Equation 4 could be approximated as

$$\mathbb{E}_{p(\mathbf{x})}\, \mathbb{E}_{p(\mathbf{y}|\mathbf{x})}\, [\mathcal{L}(\mathbf{x}, \mathbf{y})] \approx \mathbb{E}_{p(\mathbf{x})}\left[\frac{1}{L_c}\sum_{i=0}^{L_c-1}\mathcal{L}(\mathbf{x}, \widetilde{\mathbf{y}}_{n-i})\right] \tag{7}$$

where $(\widetilde{\mathbf{y}}_{n-L_c+1}, \ldots, \widetilde{\mathbf{y}}_n)$ are the $L_c$ most recent samples in the Markov chain. This approach to distribution sampling – Markov chain Monte Carlo (MCMC) sampling – is considered the gold standard in parameter inference.

In this work, we consider the Metropolis-Hastings (MH) algorithm [5, 6]. In the context of Bayesian inference for recovering labels $\mathbf{y}$ from data $\mathbf{x}$, MH sampling begins with a prior distribution $p(\mathbf{y})$, a likelihood function $p(\mathbf{x}|\mathbf{y})$, and a proposal distribution $Q(\mathbf{y}'|\mathbf{y})$ which quantifies the probability of transitioning from $\mathbf{y}$ to $\mathbf{y}'$ in the space of possible labels. Given a sample $\mathbf{y}_n$ of a Markov chain, the MH update rule is given by

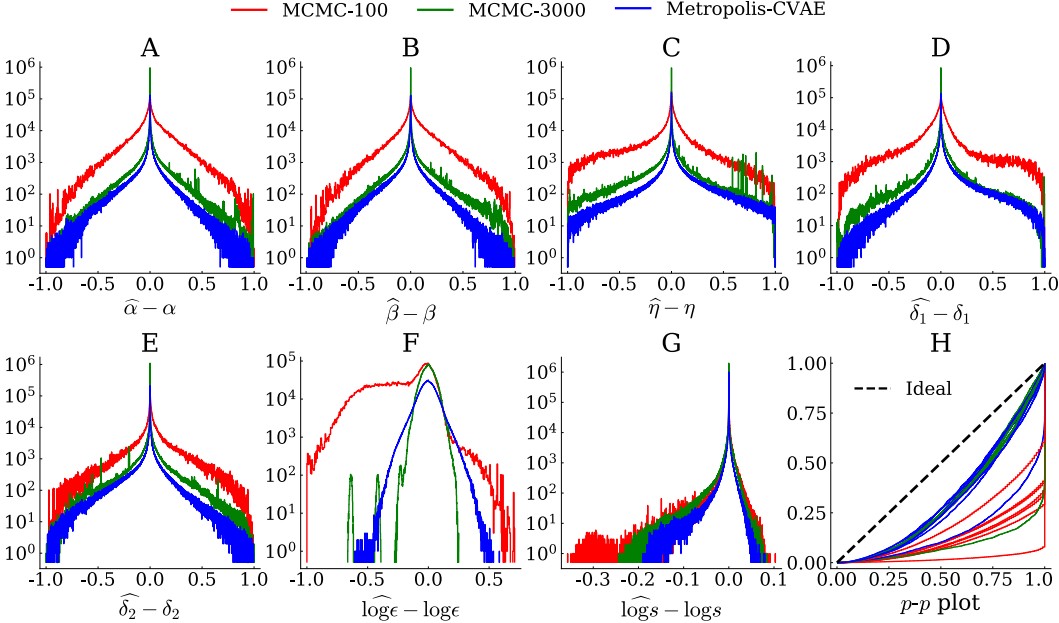

Figure 1: (A-G) Comparison of inference results for data simulated using Equation 14 with parameters drawn from Equation 17. Histograms of errors between the means of the empirical distributions and the true labels are shown. (H) $p$-$p$ plot for each label. In all plots, MCMC with 100 samples, MCMC with 3000 samples, and Metropolis-CVAE with 100 samples are shown in red, green, and blue, respectively.

$$\mathbf{y}' \sim Q(\mathbf{y}'|\mathbf{y}_n)$$
$$\alpha = \min\left(1, \frac{p(\mathbf{y}')}{p(\mathbf{y}_n)} \cdot \frac{p(\mathbf{x}|\mathbf{y}')}{p(\mathbf{x}|\mathbf{y}_n)}\right)$$
$$u \sim \mathrm{Uniform}(0, 1)$$
$$\mathbf{y}_{n+1} = \begin{cases} \mathbf{y}' & u \le \alpha \\ \mathbf{y}_n & \text{otherwise.} \end{cases} \tag{8}$$

113 The proposal distribution $Q(\mathbf{y}'|\mathbf{y})$ is a free parameter of the MH algorithm. In general, choosing $Q$
114 to resemble the true posterior as closely as possible improves the efficiency of the MH algorithm.
115 Therefore, we propose to use the approximate posterior $\hat{p}(\mathbf{y}|\mathbf{x})$ of the CVAE itself as the proposal
116 distribution. Recalling that $\hat{p}(\mathbf{y}|\mathbf{x}) = \int_{\mathbf{z}} d\mathbf{z}\, \hat{p}(\mathbf{z}|\mathbf{x})\hat{p}(\mathbf{y}|\mathbf{z}, \mathbf{x}) := \int_{\mathbf{z}} d\mathbf{z}\, E_1(\mathbf{z}|\mathbf{x})D(\mathbf{y}|\mathbf{z}, \mathbf{x})$, we let

$$Q(\mathbf{y}'|\mathbf{y}_n) = Q(\mathbf{y}') = \int_{\mathbf{z}} d\mathbf{z}\, E_1(\mathbf{z}|\mathbf{x})D(\mathbf{y}'|\mathbf{z}, \mathbf{x}) \tag{9}$$

$$\approx \frac{1}{L_{\mathbf{z}}} \sum_{i=1}^{L_{\mathbf{z}}} D(\mathbf{y}'|\mathbf{z}_i, \mathbf{x}) \quad \text{where} \quad \mathbf{z}_i \sim E_1(\mathbf{z}|\mathbf{x}). \tag{10}$$

117 By choosing this proposal function, we are able to bootstrap pseudo labels $\widetilde{\mathbf{y}}$ onto unlabeled data $\mathbf{x}_u$.
118 In particular, we minimize the semi-supervised hybrid loss

$$\mathcal{L}_{\mathrm{hybrid}} = \mathcal{L}_{\mathrm{super}} + \mathcal{L}_{\mathrm{self}} \tag{11}$$
$$\mathcal{L}_{\mathrm{super}} = \mathbb{E}_{(\mathbf{x}_\ell, \mathbf{y}) \sim \widetilde{p}_\ell(\mathbf{x}, \mathbf{y})} \left[ \mathcal{L}(\mathbf{x}_\ell, \mathbf{y}) \right] \tag{12}$$
$$\mathcal{L}_{\mathrm{self}} = \mathbb{E}_{(\mathbf{x}_u, \widetilde{\mathbf{y}}) \sim \widetilde{p}_u(\mathbf{x}, \widetilde{\mathbf{y}})} \left[ \mathcal{L}(\mathbf{x}_u, \widetilde{\mathbf{y}}) \right] \tag{13}$$

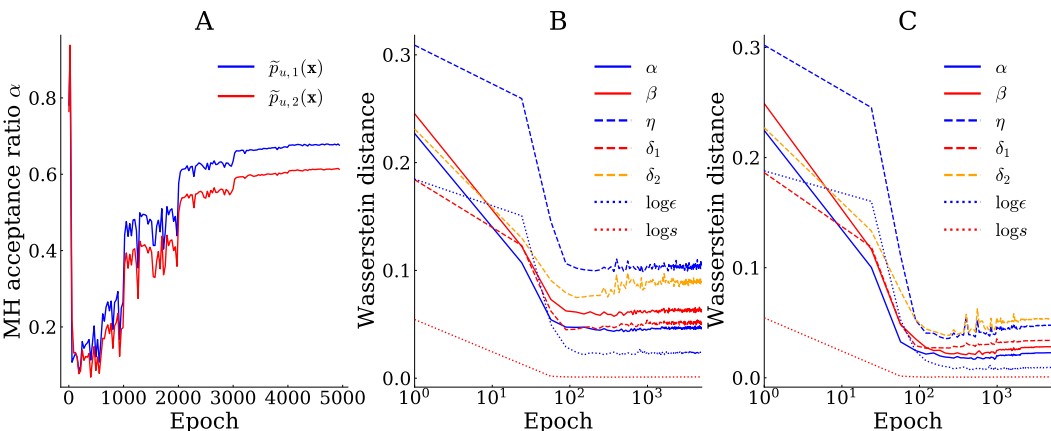

Figure 2: (A) Pseudo label acceptance rate vs. epochs for the MRI datasets $\widetilde{p}_{\ell,1}(\mathbf{x})$ and $\widetilde{p}_{\ell,2}(\mathbf{x})$. (B) Wasserstein distance vs. epochs between empirical label distributions and MCMC with 100 posterior samples, and (C) same as (B) but MCMC with 3000 posterior samples.

where $\widetilde{p}_\ell(\mathbf{x}, \mathbf{y})$ and $\widetilde{p}_u(\mathbf{x}, \widetilde{\mathbf{y}})$ are the empirical distributions over the pairs of labeled data $(\mathbf{x}_\ell, \mathbf{y})$ and unlabeled data with bootstrapped pseudo labels $(\mathbf{x}_u, \widetilde{\mathbf{y}})$, respectively. The pseudolabels $\widetilde{\mathbf{y}}$ are initialized uniformly from the prior space $p(\mathbf{y})$ and are updated according to the MH algorithm 8 at each training iteration.

Note that in the MH update step, the MH acceptance ratio $\alpha$ can be interepreted as computing a Bayesian goodness of fit check relative to the current label $\widetilde{\mathbf{y}}_n$. Therefore, $\widetilde{p}_\ell(\mathbf{x})$ and $\widetilde{p}_u(\mathbf{x})$ need not be identical distributions, merely close enough such that decreasing the supervised loss $\mathcal{L}_{\text{super}}$ improves the proposal quality for the self-supervised loss $\mathcal{L}_{\text{self}}$ early in training.

## 2.3 MRI physics application

We consider an application in magnetic resonance imaging. In MRI, data is typically acquired in the form of (3+1)D spatio-temporal grids, with 1D magnetic resonance time signals measured in each voxel of the three spatial dimensions. Advanced imaging methods typically involve voxelwise parameter inference for each time signal for the computation of quantitative maps. Modeling the individual time signals for inference, however, is challenging due to imperfections in the magnetic field generated by the scanner and other sources of signal corruption. Effectively, in MRI there is a distributional mismatch problem: simulated data $\widetilde{p}_\ell(\mathbf{x})$, with labels corresponding to well understood physics parameters, does not contain the full distribution of measured data $\widetilde{p}_u(\mathbf{x})$ which, while generated by the same physics in principle, depends on additional nuissance parameters which cannot be modeled. Machine learning models which intended to generalize to $\widetilde{p}_u(\mathbf{x})$ should therefore not be trained only on data from $\widetilde{p}_\ell(\mathbf{x})$.

In this work, we acquire multi spin-echo (MSE) MRI images. The MSE time signals are modeled using a two-component extended phase graph (EPG) model using the algorithm detailed in Prasloski et al. [7]. Using this model, the $j$-th time point for each signal is given by

$$\widehat{\mathbf{x}}_j = f(j \cdot \text{TE})$$
$$f(t) = \sum_{\ell=1}^{2} A_\ell \, \text{EPG}(t, \alpha, \beta, T_{2,\ell}, T_1) \tag{14}$$

where $\alpha$ is the spin flip angle, $\beta$ the refocusing control angle, $A_1, A_2$ the component amplitudes, $T_{2,1} \leq T_{2,2}$ the *short* and *long* transverse relaxation times, and $T_1$ the longitudinal relaxation time. The $\text{EPG}(t, \dots)$ terms are approximately exponentially decaying in $t$ with time constants $T_{2,\ell}$, with additional modifications due to MRI physics determined by $\alpha$, $\beta$, and $T_1$. The echo time TE is the uniform spacing between time points. We reparameterize $A_1, A_2, T_{2,1}$, and $T_{2,2}$ in terms of the unconstrained parameters $\eta, \delta_1$, and $\delta_2$ as follows: $A_1 = \eta$, $A_2 = 1 - \eta$, $\log T_{2,1} = \log T_{2,\min} +$

$(\log T_{2,\max} - \log T_{2,\min}) \cdot \delta_1$, and $\log T_{2,2} = \log T_{2,\min} + (\log T_{2,\max} - \log T_{2,\min}) \cdot (\delta_1 + \delta_2 \cdot (1 - \delta_1))$, where $T_{2,\min} = 10\,\text{ms}$ and $T_{2,\max} = 1\,\text{s}$. The longitudinal relaxation time is fixed $T_1 = 1.0\,\text{s}$.

MRI signal noise can be modeled as Rician [8, 9]. Given data $\mathbf{x}$ normalized to have maximum value 1, the likelihood $p(\mathbf{x}|\mathbf{y})$ under Rician noise is given by

$$\log p(\mathbf{x}|\mathbf{y}) = \sum_{j=1}^{N_{\mathbf{x}}} \log p_{\text{Rice}} \left( \mathbf{x}_j \ \middle| \ \frac{s \cdot \hat{\mathbf{x}}(\theta)_j}{\max_k \hat{\mathbf{x}}(\theta)_k} \ , \ s \cdot \epsilon \right), \tag{15}$$

$$\text{where} \qquad p_{\text{Rice}}(\xi|\nu,\sigma) = \frac{\xi}{\sigma^2} \exp\left( -\frac{\xi^2 + \nu^2}{2\sigma^2} \right) I_0 \left( \frac{\xi\nu}{\sigma^2} \right) \tag{16}$$

is the Rician probability density function with location parameter $\nu$ and scale parameter $\sigma$; $I_0$ is the modified Bessel function of the first kind with order zero. We have introduced two additional parameters: a scale parameter $s$ to account for signal normalization, and a noise level $\epsilon$ relative to this scale. Additionally, we denote $\theta = (\alpha, \beta, \eta, \delta_1, \delta_2)$ the parameters of the EPG model 14.

In total, there are 7 labels to be inferred: $\mathbf{y} = (\alpha, \beta, \eta, \delta_1, \delta_2, \log \epsilon, \log s)$. We place the following priors on the parameters:

$$
\begin{aligned}
\alpha &\sim \mathcal{TN}(180°, 45°, 90°, 180°) \\
\beta &\sim \mathcal{TN}(180°, 45°, 90°, 180°) \\
\eta &\sim \mathcal{TN}(0.0, 0.5, 0.0, 1.0) \\
\delta_1 &\sim \mathcal{TN}(0.0, 0.5, 0.0, 1.0)
\end{aligned}
\qquad
\begin{aligned}
\delta_2 &\sim \mathcal{TN}(1.0, 0.5, 0.0, 1.0) \\
\log \epsilon &\sim \mathcal{U}(\log 10^{-5}, \log 10^{-1}) \\
\log s &\sim \mathcal{TN}(0.0, 0.5, -2.5, 2.5)
\end{aligned}
\tag{17}
$$

where $\mathcal{TN}(\mu, \sigma, a, b)$ is the normal distribution with parameters $(\mu, \sigma)$ truncated to the interval $[a, b]$, and $\mathcal{U}(a, b)$ is the uniform distribution on $[a, b]$. The priors were chosen to align with the expectations that: $\alpha, \beta$ are typically near $180°$; the short component amplitude $\eta$ is typically less than the long amplitude $1 - \eta$; $\delta_1$ and $\delta_2$ should prefer to represent the shortest and longest components; the noise level $\epsilon$ is chosen uniformly from signal-to-noise ratios between 20 and 100; the scale parameter $s$ should prefer to be 1.

## 3 Experiments

### 3.1 Data sets

**MRI data** The MRI data used for this study consists of two anonymized brain scans acquired using a Carr-Purcell-Meiboom-Gill (CPMG) [10, 11] multi spin-echo sequence [12]. The first data set, denoted $\widetilde{p}_{u,1}(\mathbf{x})$, contains signals with $N_{\mathbf{x},1} = 48$ samples at times $t_i = i \cdot \text{TE}$, with echo spacing $\text{TE} = 8\,\text{ms}$, repetition time $\text{TR} = 1073\,\text{ms}$, matrix size $240 \times 240 \times 48$, and spatial resolution $0.96 \times 0.96 \times 2.5\,\text{mm}^3$. The second data set, denoted $\widetilde{p}_{u,2}(\mathbf{x})$, contains signals with $N_{\mathbf{x},2} = 56$ samples at times $t_i = i \cdot \text{TE}$, with echo spacing $\text{TE} = 7\,\text{ms}$, repetition time $\text{TR} = 1066\,\text{ms}$, matrix size $240 \times 240 \times 113$, and spatial resolution $1.0 \times 1.0 \times 3.0\,\text{mm}^3$. Following the extraction of image volumes containing the brain, $\widetilde{p}_{u,1}(\mathbf{x})$ and $\widetilde{p}_{u,2}(\mathbf{x})$ contain 821 145 and 1 265 306 signals, respectively. MRI data was acquired on a $3\,\text{T}$ MR system (Ingenia Elition, Philips Medical Systems, Best, The Netherlands) from healthy volunteers giving written and informed consent, and approved by our university ethics board.

**Simulated data** Using the EPG physics model 14 with Rician noise, we consider two simulated data sets. First, the labeled data set $\widetilde{p}_{\ell}(\mathbf{x})$ which is generated on demand during training using labels $\mathbf{y} \sim p(\mathbf{y})$ drawn from the prior distributions 17. Second, a precomputed simulated data set $\widetilde{p}_{u,3}(\mathbf{x})$ used for validation of the method, where the labels are held out during training. All simulated signals are generated with $N_{\mathbf{x},3} = 64$ samples and $\text{TE} = 10\,\text{ms}$.

**MCMC data** MCMC is performed using the No-U-Turn Sampler [13] algorithm to generate posterior samples $\hat{\mathbf{y}} \sim p(\mathbf{y}|\mathbf{x}_u)$ which can be compared with the (Metropolis-)CVAE posterior

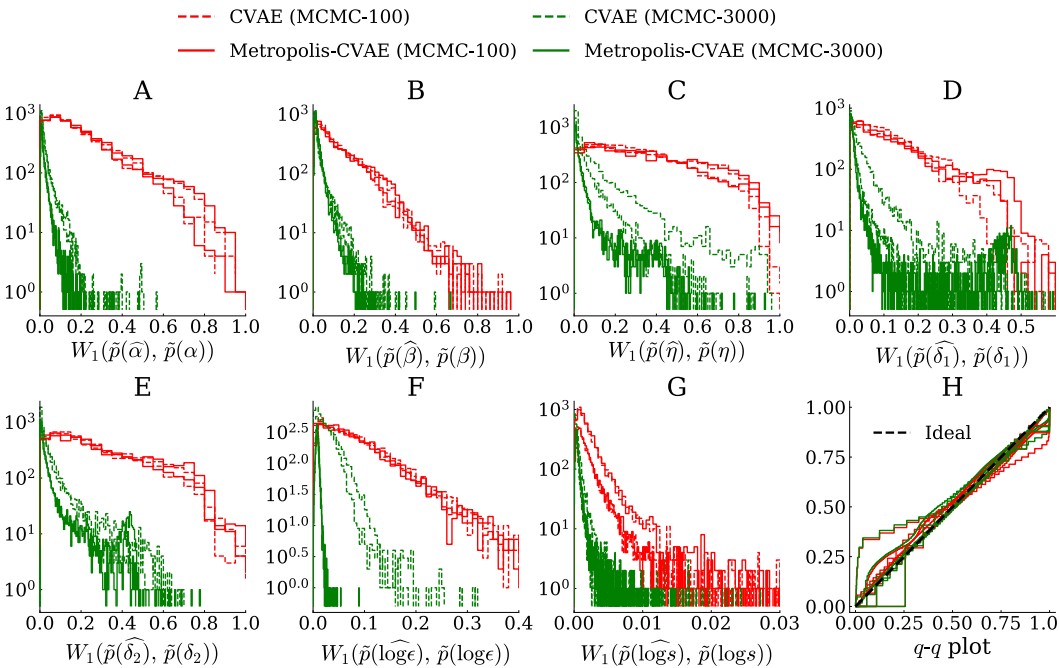

Figure 3: (A-G) Histograms of Wasserstein distances between (Metropolis-)CVAE and MCMC empirical distributions for the two MRI data sets. Histograms of errors between the means of the empirical distributions and the true labels are shown. (H) Quantile-quantile plot for each label between global empirical distributions of CVAE samples and MCMC samples. In all plots, comparison with MCMC with 100 samples and 3000 samples are shown in red and green, respectively; CVAE trained only on simulated data is shown in dashed lines; Metropolis-CVAE shown in solid lines.

samples. MCMC was performed twice: once to draw 100 posterior samples for every time signal in $\widetilde{p}_{u,1}(\mathbf{x})$, $\widetilde{p}_{u,2}(\mathbf{x})$, and $\widetilde{p}_{u,3}(\mathbf{x})$, and once to draw 3000 posterior samples for a subset of 5000 signals from the training, validation, and testing partitions of all three data sets. In total, performing the above MCMC analysis took approximately $72\,\mathrm{h}$ using an AMD Ryzen 9 3950X 16-Core CPU.

## 3.2 Model architecture

**CVAE components**   Let $\mathbf{x} \in \mathbb{R}^{N_\mathbf{x}}$, $\mathbf{y} \in \mathbb{R}^{N_\mathbf{y}}$, and $\mathbf{z} \in \mathbb{R}^{N_\mathbf{z}}$ be input data, corresponding labels, and latent space samples, respectively. The encoders $E_1$ and $E_2$ are chosen to be multivariate normal distributions: $E_1(\mathbf{z}|\mathbf{x}) = \mathcal{N}(\mu_{\mathbf{z}_1}, \sigma_{\mathbf{z}_1})$ and $E_2(\mathbf{z}|\mathbf{x}, \mathbf{y}) = \mathcal{N}(\mu_{\mathbf{z}_2}, \sigma_{\mathbf{z}_2})$, where $\mu_{\mathbf{z}_1}, \mu_{\mathbf{z}_2} \in \mathbb{R}^{N_\mathbf{z}}$ and $\sigma_{\mathbf{z}_1}, \sigma_{\mathbf{z}_2} \in \mathbb{R}^{N_\mathbf{z}}_+$. Similarly, the decoder is given by $D(\mathbf{y}|\mathbf{x}, \mathbf{z}) = \mathcal{TN}(\mu_\mathbf{y}, \sigma_\mathbf{y}, 0, 1)$, where $\mu_\mathbf{y} \in \mathbb{R}^{N_\mathbf{y}}$ and $\sigma_\mathbf{y} \in \mathbb{R}^{N_\mathbf{y}}_+$ parameterize independent multivariate normal distributions truncated to $[0, 1]$. The labels $\mathbf{y}$ are scaled linearly from the prior domains 17 to $[0, 1]^{N_\mathbf{y}}$ in order to better condition the network during training. Similarly, as we are interested only in the relative $\mathbf{x}$ values and not their absolute scale, inputs $\mathbf{x}$ are normalized to $[0, 1]^{N_\mathbf{x}}$.

Each of the $E_1$, $E_2$, and $D$ networks are composed of fully connected layers with $\mathrm{ReLU}$ activation functions and $H = 2$ hidden layers, with hidden dimension $N_H = 512$, for a total of $H + 2 = 4$ layers. The dimensions of the data, labels, and latent space are $N_\mathbf{x} = 64$, $N_\mathbf{y} = 7$, and $N_\mathbf{z} = 12$, respectively. The encoder networks output $\mu_{\mathbf{z}_1}$, $\log \sigma_{\mathbf{z}_1}$, $\mu_{\mathbf{z}_2}$, and $\log \sigma_{\mathbf{z}_2}$ vectors. In order to avoid latent space collapse during early training stages – that is, one or both of $|\mu_{\mathbf{z}_i}|, |\log \sigma_{\mathbf{z}_i}| \to \infty$ – each $\mu_{\mathbf{z}_i}$ was bounded to $(-3, 3)$ using the activation function $x \to 3 \tanh(x)$, and each $\log \sigma_{\mathbf{z}_i}$ was bounded to $(-6, 0)$ using the activation function $x \to 3 \tanh(x) - 3$. These bounds were determined by observing empirical $\mu_{\mathbf{z}_i}$ and $\log \sigma_{\mathbf{z}_i}$ values and choosing intervals which clipped only the tails of the distributions. The decoder network outputs $\mu_\mathbf{y}$ and $\log \sigma_\mathbf{y}$ vectors without further nonlinearities.

**Loss functions** The terms $\mathcal{L}_{\text{super}}$ and $\mathcal{L}_{\text{self}}$ in Equation 11 are identical apart from their inputs, with each term consisting of the KL-divergence and evidence lower bound (ELBO) terms from the variational lower bound of Equation 1. The KL-divergence can be computed in closed closed-form, and is given by

$$\text{KL}\left(E_2(\mathbf{z}|\mathbf{x},\mathbf{y}) \,\|\, E_1(\mathbf{z}|\mathbf{x})\right) = \sum_{j=1}^{N_\mathbf{z}} \frac{\sigma_{\mathbf{z}_2,j}^2 + (\mu_{\mathbf{z}_2,j} - \mu_{\mathbf{z}_1,j})^2}{2\sigma_{\mathbf{z}_1,j}^2} + \log\frac{\sigma_{\mathbf{z}_1,j}}{\sigma_{\mathbf{z}_2,j}} - \frac{1}{2} \tag{18}$$

Following sampling $\mathbf{z}_2 \sim E_2(\mathbf{z}|\mathbf{x},\mathbf{y})$ and subsequently computing $(\mu_\mathbf{y}, \log\sigma_\mathbf{y}) = D(\mathbf{y}|\mathbf{x},\mathbf{z}_2)$, the ELBO component is approximated as

$$\mathbb{E}_{E_2(\mathbf{z}|\mathbf{x},\mathbf{y})}\left[\log D(\mathbf{y}|\mathbf{x},\mathbf{z})\right] \approx \sum_{j=1}^{N_\mathbf{y}} \Bigg\{$$
$$\log\phi\left(\frac{\mathbf{y}_j - \mu_{\mathbf{y},j}}{\sigma_{\mathbf{y},j}}\right) - \log\left(\Phi\left(\frac{1 - \mu_{\mathbf{y},j}}{\sigma_{\mathbf{y},j}}\right) - \Phi\left(\frac{0 - \mu_{\mathbf{y},j}}{\sigma_{\mathbf{y},j}}\right)\right) - \log\sigma_{\mathbf{y},j}\Bigg\}$$
$$\text{where} \qquad \phi(\xi) = \frac{1}{\sqrt{2\pi}}\exp\left(-\frac{1}{2}\xi^2\right) \tag{19}$$
$$\Phi(\zeta) = \frac{1}{2}\left(1 + \text{erf}\left(\frac{\zeta}{\sqrt{2}}\right)\right).$$

$\phi(\xi)$ and $\Phi(\zeta)$ are the probability density and cumulative distribution functions of the standard normal distribution, respectively. Note that each term in the sum of Equation 19 is the log-likelihood of a normal distribution with parameters $(\mu_{\mathbf{y},i}, \sigma_{\mathbf{y},i})$ truncated to the unit interval $[0,1]$.

### 3.3 Training

All data sets were split into training/validation/testing with proportions $50\,\%/25\,\%/25\,\%$. Input data was padded with zeros to length $N_\mathbf{x} = 64$, if necessary. To support data with differing unpadded lengths, random masking was performed during training: for each $\mathbf{x}$, a random integer $j_{\text{mask}}$ was sampled from 32–64 and all elements $\mathbf{x}_{j>j_{\text{mask}}}$ were set to zero. The ADAM optimizer was used with an initial learning rate of $10^{-4}$. The learning rate was decreased every 1000 epochs by a factor of $\sqrt{10}$. Training was completed after 5000 epochs, where an epoch is defined as 100 iterations. Each iteration, a batch of 1024 data and (pseudo-)label pairs are drawn from either a labeled or an unlabeled dataset, with equal probability. If labeled data are sampled, the loss component $\mathcal{L}_{\text{super}}(\mathbf{x}_\ell, \mathbf{y})$ is descended on. If unlabeled data are sampled, the pseudo labels $\widetilde{\mathbf{y}}$ corresponding to the sampled $\mathbf{x}_u$ are updated using Equation 8 before descending on the loss component $\mathcal{L}_{\text{self}}(\mathbf{x}_u, \widetilde{\mathbf{y}})$. In the Metropolis-Hastings update step, we set $L_c = 1$ in equation 7 and $L_\mathbf{z} = 1$ in equation 9. Training was performed on a single Nvidia GeForce RTX 3080 GPU with 10 GB of VRAM; approximately 15 hours was required to train for 5000 epochs.

## 4 Results and discussion

In the first experiment, a Metropolis-CVAE is trained on online simulated labeled data $\mathbf{x}_\ell \sim \widetilde{p}_\ell(\mathbf{x})$ as well as precomputed simulated data $\mathbf{x}_u \sim \widetilde{p}_{u,3}(\mathbf{x})$ with labels held out during training. Figure 1(A-G) shows the distributions of prediction errors by the trained network for each label. For each method, the prediction error is defined as the difference between the true label and the mean of the posterior samples. The labels have been normalized to $[0,1]$. The histograms for the Metropolis-CVAE with 100 posterior samples are more tightly clustered around zero than MCMC with either 100 or 3000 posterior samples – denoted MCMC-100 and MCMC-3000 – in all cases except for one (Figure 1F). Figure 1H shows a $p$-$p$ plot: the fraction of posterior samples greater than or equal to the true label, $p$, is plotted against the cumulative distribution of $p$-values across the data set; the Metropolis-CVAE produces similar curves as MCMC-3000.

In the second experiment, a Metropolis-CVAE is similarly trained using labeled simulated $\mathbf{x}_\ell \sim \widetilde{p}_\ell(\mathbf{x})$, but now with the unlabeled data $\mathbf{x}_u$ drawn from the MRI data sets $\widetilde{p}_{u,1}(\mathbf{x})$ and $\widetilde{p}_{u,2}(\mathbf{x})$. For comparison, a second traditional CVAE is trained on the simulated data only. Figure 2A shows the acceptance rate of the proposed pseudo labels $\widetilde{\mathbf{y}}' \sim Q(\mathbf{y}')$ for each data set during training. As the Metropolis-CVAE continually learns from both data sets, we find that the network quickly enters a negative feedback loop in which the acceptance rate continually increases throughout training. Noteworthy is that the acceptance rate of $\widetilde{\mathbf{y}}$ converges to a value between 0.6 and 0.7 for both datasets. The default target acceptance rate of the No-U-Turn sampler is 0.65, a value which originates from a theoretical result pertaining to Hamiltonian Monte Carlo (HMC) [14]. Theoretical work would be required to demonstrate a direct connection between Metropolis-CVAEs and the HMC result. However, this illustrates that the network neither accepts nor rejects more proposals than is typically desired. Figure 2(B-C) shows the average Wasserstein distance between empirical label distributions from the Metropolis-CVAE with MCMC-100 and MCMC-3000, respectively. During training, the Metropolis-CVAE quickly reaches minimum Wasserstein distances with respect to MCMC-100. Minima are reached with respect to MCMC-3000 similarly quickly. We can extrapolate from this result that the Metropolis-CVAE likely produces higher quality posterior samples than MCMC-3000. This may be expected, as the Metropolis-CVAE updates its pseudo labels at every training iteration, continuously performing MCMC throughout training using the MH update rule 8.

Figure 3(A-G) compares the empirical distributions of a traditional CVAE trained only on simulated data with a Metropolis-CVAE. Histograms of the Wasserstein distances between the two networks and MCMC-100 do not show large differences, due to the low-quality of MCMC-100. Compared with MCMC-3000, parameters which are traditionally easier to infer, such as $\alpha$ and $\beta$, show little difference between the networks. The CVAE trained on supervised data alone shows significant deviation from MCMC-3000 for several parameters, particularly $\eta$, $\delta_1$, and $\log \epsilon$. This illustrates the ability of the Metropolis-CVAE to generalize well to the unlabeled test data, as unlabeled training data from $\widetilde{p}_u(\mathbf{x})$ has been explicitly incorporated into its training process. Figure 3(H) shows quantile-quantile plots for distributions of all label samples across the data sets. All methods agree.

**Limitations**    A limitation of this method is the requirement of a likelihood function which is fast to compute in order to perform the Metropolis-Hastings update step 8. For physics models, computing the likelihood of a set of parameters often involves costly forward simulations of complex models. Therefore, computationally intensive models which require solving non-trivial integral or differential equations will not be suitable.

Another limitation is that the pseudo label sampling procedure may converge to a stable local minimum which is far from the globally optimal labels. This is inherent to the MH update step; while MCMC methods often provide convergence guarantees in the limit of large numbers of update steps, one can not predict how many update steps this will require in practice. However, by training the Metropolis-CVAE on both simulated data generated from the prior space as well as real unlabeled data, we have not found this to be a practical concern.

**Potential negative societal impacts**    This work describes a framework for ascribing labels to unlabeled data. Potential malicious and unintended uses could occur if this framework were significantly extended beyond the MRI physics inference problem which we considered. For example, this methodology could be used to infer missing data generally, with the inferred data then presented under the pretense that it were true data. Further, we have not shown that our method of data inference for unlabeled data is inherently fair, nor that the clinical MRI data under which the Metropolis-CVAE model is trained on cannot be recovered from the network weights.

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
