# OpenReview forum: "Metropolis-CVAE: Bootstrapping Labels for Bayesian Inference via Semi-Supervised Conditional Variational Autoencoders"
_NeurIPS.cc/2021/Conference — NeurIPS 2021 Submitted_

### Official Review · Reviewer_J7Vm · 2021-07-14

**Rating:** 6
**Confidence:** 2

**Summary:**

The authors present a scheme for parameter inference that builds on both simulated/labelled data and real/unlabelled data.
It uses MCMC with a CVAE proposal distribution to infer the data labels, and then trains the CVAE on the mixed data, bootstrapping the process.
They evaluate their method on an MRI task.

**Limitations And Societal Impact:**

These were adequately addressed.

**Main Review:**

# Overall review

I really liked this paper and have tentatively recommended acceptance.
The main caveat in my recommendation is that I couldn't fully read Figure 3 - I have asked for clarification below.
Moreover, because I like what you've done so much, it feels like a shame that you haven't done more to explore the opportunities of the algorithm more generally.
I've left some thoughts on this below.
Whether or not the paper is accepted in its current form, I would encourage you to explore this a bit further in order to increase the impact and significance of the work (I think you have a little extra space in the paper that could be used for this).

# Major comments

Your main contribution seems to be a strategy for inferring parameters for generative models in a label-efficient way.
I like your overall strategy.
I am mindful of the fact that I think my research background is slightly different than yours (just inferring from the paper), so some of our assumed context may differ.
I've tried to account for this in my review, but we should be extra careful not to accidentally talk past each other, and if I seem to have misunderstood your points then please help me understand.

I don't have the background to know if your interpretation of MRI imaging as mixing simulated labelled data with unlabelled real data is novel.
If it is you should make this more explicit, and if not you should cite the source.

I cannot comment on the correctness of the work between lines 139 and 163.

Personally, I am most interested in the algorithmic contribution: the mixing of MCMC pseudo-data alongside simulated labelled data.
Here, it feels like there are some design options that could benefit from more investigation and discussion.
- What are the trade-offs in performing more than one MCMC step for each new label?
- What heuristics should be used to balance the gradient-based optimization of the proposal distribution/inferred parameters against the MCMC label inference?
- Are there advantages to 'warm-starting' the CVAE proposal in various ways?
- Why do you randomly select labelled or unlabelled points rather than using a steady ratio between them? What might the best ratio be? Does this change during training?

It seems as though the impact of this work is much greater if the method can be generalized to other settings and data-sets.
I think I'm sufficiently interested in the method already to tentatively recommend acceptance, but I think you could get much more out of this with more exploration of the algorithm itself.

Certainly, the paper would benefit from an algorithm box.
If nothing else, this might force you to enumerate the design assumptions that you make in implementing the algorithm, and give you an opportunity to discuss alternative choices.

Can you also say a little more about the hyperparameter choices made in S3.3? A number of these choices seem quite specific.
If you're just using something from a previous implementation that's absolutely fine, but it's nice to know what the origin of the choices is.

This comment is the most important for me: I cannot read Figure 3 and it seems the most important figure in the paper.
This is your opportunity to show that adding the unsupervised data improves your model, which is your main contribution.
I think this could be improved just by switching colors---unfortunately the line-shape makes it very hard to distinguish dashed lines from full ones.
I'm going to give benefit of the doubt for the moment that the results show the Metropolis-CVAE helping on at least a few of these.
If it is at all possible, please replace this figure with a more legible one during the review stage, as my provisional score is sensitive to this graph showing sufficient difference between the two methods.

If you can think of anything further that can be done to highlight performance differences between the CVAE and Metropolis-CVAE, these would be most likely to reaffirm my belief that this paper should be accepted.

# Minor remarks

The abstract could be much shorter.

Does the use of two different data sources for the MRI data complicate things?
How do you deal with the mismatch?
(Other than the masking - after all the voxels are different sizes etc.)

I usually don't work with VAEs, and I understand that truncating the Gaussian is more usual in that domain.
It does, however, mean that the KL divergence estimate in equation (18) is wrong because the approximate posterior which is sampled from is not a Gaussian.
I know most people ignore this, so I don't hold it against this paper, but it seems worth noting.
Can I just check also whether you truncate by resampling or by clamping values greater than 1 down to 1?
These are quite different distributions.

Computationally, checking each step whether you are using one loss or the other, then waiting for your CPU to return the pseudo-label, and then doing a backwards pass, is very bad (removes most of the advantage of GPU by blocking parallelization).
I think it's beyond the scope of this work, but it would be valuable to consider possible parallelization strategies or ways to pre-compute the pseudo-labels in batches.

**Time Spent Reviewing:**

2

---

> ### Author Response · Authors · 2021-08-11
> **Initial response to Reviewer 4 (J7Vm)**
>
> **Reviewer 4.1:** Main Review: Overall review [...]
>
> **Response 4.1:** We agree. Please see our comment addressed to all reviewers.
>
> **Reviewer 4.2:** I don't have the background to know if your interpretation of MRI imaging as mixing simulated labelled data with unlabelled real data is novel. [...]
>
> **Response 4.2:** To the best of our knowledge, this mixture of simulated and unlabeled data is novel in MRI. Generally speaking, ML models trained for MRI applications use supervised approaches using (often highly sophisticated) physics simulations. However, the distributional mismatch between training and inference is thenceforth disregarded. We will clarify this in the manuscript.
>
> **Reviewer 4.3:** Personally, I am most interested in the algorithmic contribution: the mixing of MCMC pseudo-data alongside simulated labelled data. [...]
>
> **Response 4.3:** We thoroughly agree that these avenues ought to be investigated in this work, at the very least as supplementary material. In fact, we did perform cursory explorations into CVAE warm-starting. For example, maximum likelihood estimation- and MCMC-generated labels were used as both: (i) targets for the unlabeled data for supervised pretraining, and (ii) initial pseudo labels for the Metropolis-CVAE. We found that while the time to convergence for the Metropolis-CVAE decreased, the final performance was unchanged.
>
> **Reviewer 4.4:** It seems as though the impact of this work is much greater if the method can be generalized to other settings and data-sets. [...]
>
> **Response 4.4:** We agree. Please see our comment addressed to all reviewers.
>
> **Reviewer 4.5:** Certainly, the paper would benefit from an algorithm box. [...]
>
> **Response 4.5:** We agree.
>
> **Reviewer 4.6:** Can you also say a little more about the hyperparameter choices made in S3.3? [...]
>
> **Response 4.6:** Hyperparameter choices were not deliberately specific. The choice of $N_x = 64$, as well as the signal masking to lengths between $32$ and $64$, corresponds to the expected minimum and maximum lengths of acquired MRI time signals in our application domain. Alternate optimizers to ADAM were not investigated. The learning rate schedule was not thoroughly investigated, besides two important findings: initial learning rate much larger than 1e-4 does not help training, and can lead to instabilities; frequently dropping the learning rate increases performance. Varying the total training epochs was not investigated, beyond confirming that 5000 epochs was sufficient for convergence in all cases. Batch size 1024 was not investigated beyond the observation that, given such small data of only 64 elements, too small of a batch size quickly becomes extremely inefficient at utilizing the GPU parallelism. The choice to randomly select a loss function to descend on, compared to a sum of losses or similar, was also not investigated. Choosing $L_c = L_z = 1$ was also not investigated, in order to minimize the number of hyperparameters.
>
> **Reviewer 4.7:** This comment is the most important for me: I cannot read Figure 3 [...]
>
> **Response 4.7:** We apologize for the lack of clarity of the figure; this will certainly be improved in the revision stage. We would like to clarify some possible misunderstandings about the results:
> * Due to the lack of  “gold standard” parameters for the unlabeled MRI data, a comparison between the Metropolis-CVAE and the CVAE is challenging. The Metropolis-CVAE likely produces higher quality posterior samples than the MCMC (lines 251-257 of the discussion), as the Wasserstein distances between Metropolis-CVAE and MCMC posterior samples decrease as the number of MCMC samples increases (Figure 2 B-C). That is, the MCMC posteriors approach those of the Metropolis-CVAE as the MCMC quality increases. This hypothesis is supported by Figure 1 wherein the Metropolis-CVAE outperforms MCMC for nearly all parameter posteriors.
> * In subsequent revisions, the comparisons with MCMC-100 (red lines) in Figure 3 will be removed; in hindsight, the MCMC-100 samples are too low quality to facilitate a useful “gold standard”. This will simplify the figure, and help highlight that the solid green lines (Metropolis-CVAE) lie almost entirely below the corresponding dashed green lines (CVAE), meaning that - at least compared to MCMC-3000 - the Metropolis-CVAE produces higher quality samples.
>
> **Reviewer 4.8:** If you can think of anything further that can be done to highlight performance differences between the CVAE and Metropolis-CVAE [...]
>
> **Response 4.8:** Yes, we intend to investigate Metropolis-CVAE performance as a function of distribution shift in a toy Bayesian regression problem. Additionally, we will slightly modify the first MRI experiment, introducing a distribution shift between the labeled and unlabeled simulated data. We hypothesize that these experiments will help highlight the performance improvement of the Metropolis-CVAE compared to vanilla CVAE. Please see our comment addressed to all reviewers for more details about these proposed experiments.
>
> **Reviewer 4.9:** The abstract could be much shorter.
>
> **Response 4.9:** Agreed.
>
> **Reviewer 4.10:** Does the use of two different data sources for the MRI data complicate things? [...]
>
> **Response 4.10:** The differing data sources do not complicate the situation very much, apart from the data augmentation described in Section 3.3, which is required to handle time signals of different lengths (48 vs. 56 for the data sources, and 32-64 for the simulated data). The different data sources arise from the same type of MRI scan (CPMG), and therefore the underlying MRI physics is not meaningfully changed. Rather, the data sets correspond to different regions of parameter space. For example, while the voxel size affects the time signal in terms of signal-to-noise ratio, signal noise is incorporated into the physics model in terms of the $\log\epsilon$ noise-level parameter.
>
> **Reviewer 4.11:** I usually don't work with VAEs, and I understand that truncating the Gaussian is more usual in that domain. It does, however, mean that the KL divergence estimate in equation (18) is wrong [...]
>
> **Response 4.11:** This is actually not the case; the latent space distributions are still Gaussians, and Equation (18) holds. Note that it is not the sampling from the latent distributions which is being truncated, but rather the $\mu, \log\sigma$ parameters of the Gaussian distributions which parameterize the latent spaces are being restricted to lie in finite intervals rather than $\mathbb{R}$. This is to help prevent the latent space distributions from collapsing during training, which would result in degenerate latent samples. We will clarify this section of the manuscript.
>
> **Reviewer 4.12:** Can I just check also whether you truncate by resampling or by clamping values greater than 1 down to 1? [...]
>
> **Response 4.12:** Sampling from the truncated normal distributions is performed neither by resampling nor by clamping; we use a combination of the reparameterization trick and the inverse transform method. This method is conveniently continuously differentiable with respect to the four truncated normal distribution parameters:
> * Let $u \sim U(0,1)$. Then $z = \mu + \Phi ^{-1}\left(\Phi (a)+u\cdot (\Phi (b)-\Phi (a))\right)\sigma \sim \mathcal{TN}(\mu,\sigma,a,b)$, where $\Phi$ is the cumulative distribution function (CDF) of the unit normal distribution and $\Phi^{-1}$ its inverse. See for example the Wikipedia article section “Generating values from the truncated normal distribution”: https://en.wikipedia.org/wiki/Truncated_normal_distribution#Generating_values_from_the_truncated_normal_distribution.
>
> **Reviewer 4.13:** Computationally, checking each step whether you are using one loss or the other, then waiting [...]
>
> **Response 4.13:** We agree, this is an interesting optimization to consider for future work.

---

> > ### Comment · Reviewer_J7Vm · 2021-09-01
> > **No change**
> >
> > Thank you for your response. I still basically like this paper, but after some discussion I don't feel able to increase my score further.
> >
> > Most of your specific responses above make sense and seem correct.
> >
> > A few comments though on the truncated Gaussian, which doesn't really affect my score at all, but is interesting.
> > I'm fairly sure you do have to be careful computing the KL term of the VAE loss. There is one KL expression for the divergence between two Gaussian distributions. But if one of the distributions has finite domain then the integral will have a different exact solution. In practice you can probably ignore this so long as the truncation distance is sufficiently large.

---

### Official Review · Reviewer_bwY8 · 2021-07-16

**Rating:** 6
**Confidence:** 4

**Summary:**

Variational inference and distribution sampling are two popular methods for statistical inference. The present paper combines these two approaches in a semi-supervised learning setting, where the distribution of unlabeled data may differ from that of labeled data. Experimental results are provided for an MRI dataset, where the task is to infer the imaging/tissue parameters.

**Limitations And Societal Impact:**

Yes.

**Main Review:**

Variational inference and distribution sampling are both powerful approaches, and combining them is likely to be impactful. Here, the authors present a reasonable and novel strategy for semi-supervised learning.

- Overall, I find the method interesting, but experimental justification could be significantly expanded: (i) How would other semi-supervised methods, for instance the cited Kingma et al paper, perform on this task? (ii) Could the authors add 1-2 more tasks that are studied in previous semi-supervised learning papers?
- Somewhat related to the previous point, how would this method extend to scenarios with discrete labels?
- The authors already discuss sampling-related distributions of the proposed method. But, I think it would be valuable to put this in a better context: For some of the more popular applications in ML conferences such as image/audio analysis, language processing, recommendation systems, how applicable is this approach? It'd be ok if applicability is low for many tasks, but I believe such discussion would improve the paper.
- I find the review of previous work somewhat missing. A better survey of the semi-supervised learning literature would significantly improve the better. I think the current flow converges to CVAE very quickly.
- The presentation of the material is clear.

----------------------------- UPDATE -----------------------------

I like the main idea behind this paper. I also think that the improvement plan proposed by the authors sounds reasonable and interesting. However, considering that it is certainly possible for this set of experiments to reveal no significant practical improvements with respect to baselines, I decided to keep my score at its current level.

**Time Spent Reviewing:**

7

---

> ### Author Response · Authors · 2021-08-11
> **Initial response to Reviewer 3 (bwY8)**
>
> **Reviewer 3.1:** Overall, I find the method interesting, but experimental justification could be significantly expanded: [...]
>
> **Response 3.1:** We agree. Additionally, these sentiments are generally shared by all four reviewers. Please see our comment addressed to all reviewers.
>
> **Reviewer 3.2:** Somewhat related to the previous point, how would this method extend to scenarios with discrete labels?
>
> **Response 3.2:** We have not considered discrete labels. This would need to be investigated further, but the following modification may be sufficient: for a discrete label $\mathbf{y}$ with $K$ possible values, let $\alpha = (p_1, \ldots, p_K)$ with $\sum_i p_i = 1$ parameterize a Categorical distribution with $K$ categories. Then $\alpha$ should be output from the CVAE, and the Categorical probability mass function $p(\mathbf{y}|\alpha)$ should be used for the ELBO loss component in Equation (19) with the discrete label $\mathbf{y}$, as opposed to the truncated normal probability density function $p(\mathbf{y}|\mu,\sigma,a,b)$. Finally, in the MH update step, sample $\mathbf{y}^\prime$ from the Categorical distribution and subsequently update the discrete pseudo label $\tilde{\mathbf{y}}$ according to Equation (8) as usual.
>
> **Reviewer 3.3:** The authors already discuss sampling-related distributions of the proposed method. But, I think it would be valuable to put this in a better context:
> [...]
>
> **Response 3.3:** Thank you for the suggestion - this is an interesting point which is indeed worth incorporating into the discussion. As far as the listed applications, it is not clear how such a connection would be made, though it does not seem fundamentally incompatible. In the context of physics, for example, the required likelihood function for the Metropolis-Hastings sampling is often constructed as a combination of a physics model and a measurement noise model. It is not immediately clear how an analogous approach could be used for e.g. natural language processing. However, if a likelihood function were available through some other construction, then Metropolis-CVAEs would be applicable.
>
> **Reviewer 3.4:** I find the review of previous work somewhat missing. [...]
>
> **Response 3.4:** We agree. The transition directly into CVAE literature is too abrupt; the literature review will be bolstered in subsequent revisions.

---

### Official Review · Reviewer_ba7Q · 2021-07-16

**Rating:** 4
**Confidence:** 4

**Summary:**

This paper addresses semi-supervised learning problem. The authors propose to adopt Metropolis-Hastings Markov chain Monte Carlo (MH-MCMC) into the sampling process of posterior in conditional variational autoencoder (CVAE) learning.
Experimental results on MRI datasets demonstrate that the proposed method performs better than existing methods regarding the number of samplings.

**Ethical Concerns:**

NA.

**Limitations And Societal Impact:**

They addressed the limitations and potential negative social impact of their work.

**Main Review:**

### Originality:
* This work proposes a novel combination of CVAE for semi-supervised learning and MH-MCMC.

### Quality:
* The problem setting this paper addresses is not clear.
    - The problem setting is finally described in the experiment section, so I cannot find whether it represents the scope of the proposed approach or the scope of the specific experiments conducted here.
    - In Section 2.3, it would be better to focus more on the machine learning perspective.

* The motivation to use MH-MCMC is not described in the main text.
    - The authors do not specify a clear reason/purpose to introduce this combination of CVAE for semi-supervised learning and MH-MCMC.
    - The specific combination would be novel, but there have been many combinations of other sampling methods including MCMC, and CVAE learning. Therefore, they should differentiate their approach from them.
    - In Section 2.1 (related work), it would be better to focus on comparing the proposed approach and existing works.

* In experiments, although the authors claim the efficiency of the proposed approach, they did not provide a comparison on actual computational time. Thus, it limits the significance of the analysis. (I acknowledge that they provided the analysis in terms of the number of sampling.)


### Clarity:
* The problem setting and the motivation of the proposed approach are unclear, as stated above.

* Minner comments.
    - What is the difference between $q$ in Eq.(1) and $\hat{p}$ in Eq.(2)? Both seem to represent approximate posterior.
    - In Eq.(10), $L_z$ is not defined.


### Significance:
* Unclear scope of this paper addresses degrades significance.

**Time Spent Reviewing:**

5

---

> ### Author Response · Authors · 2021-08-11
> **Initial response to Reviewer 2 (ba7Q)**
>
> **Reviewer 2.1:** The problem setting this paper addresses is not clear. [...]
>
> **Response 2.1:** We agree. The Metropolis-CVAE method is a fully general semi-supervised inference method, and this should be stressed more clearly in the text. All that is required is: (i) a set of labeled data, (ii) a set of unlabeled data for which labels are to be inferred, and (iii) a likelihood function for evaluating proposed labels given data. It was our intention that Sections 1, 2.1, and 2.2 were written largely without reference to a specific experiment in order to implicitly emphasize the generality of the method; this generality will be stated more explicitly in subsequent revisions. Please note that the scope is partially addressed in lines 40-44 of Section 1: Metropolis-CVAEs are introduced as a combination of “traditional CVAEs with the Metropolis-Hastings Markov chain monte carlo (MCMC) inference method”, both of which are general inference methods.
>
> **Reviewer 2.2:** In Section 2.3, it would be better to focus more on the machine learning perspective.
>
> **Response 2.2:** We agree. This section will be rewritten to introduce the MRI inference problem as an application of the machine learning methodology, with the MRI details moved to the supplementary material where appropriate.
>
> **Reviewer 2.3:** The motivation to use MH-MCMC is not described in the main text. [...]
>
> **Response 2.3:** We agree that our specific combination of the Metropolis-Hastings (MH) algorithm with CVAEs is not well differentiated from the set of all possible combinations of MCMC methods and CVAE learning. The justification for choosing MH as the sampling method over more advanced MCMC methods is simplicity: the MH algorithm is arguably the simplest possible MCMC method, as all that is required to accept or reject a proposed sample is a ratio of posterior probabilities; see Equation (8). This justification will be added to the manuscript.
>
> **Reviewer 2.4:** In Section 2.1 (related work), it would be better to focus on comparing the proposed approach and existing works.
>
> **Response 2.4:** This is somewhat difficult to achieve given the limited space, as in order to appreciate the contributions of the Metropolis-CVAE method and compare it with existing works, one first must recall the related work of CVAEs. Perhaps instead, it would suffice to introduce a section between Section 2.2 and 2.3 which explicitly compares and contrasts Metropolis-CVAEs with related work, with Section 2.1 renamed to “Background” or similar.
>
> **Reviewer 2.5:** In experiments, although the authors claim the efficiency of the proposed approach [...]
>
> **Response 2.5:** We do state the computation time for the MCMC analysis; please see lines 186-187. The total time to perform the MCMC analysis for this work was approximately 72 hours. We agree, however, that we do not explicitly state the time to perform inference for the CVAE (note that the inference process for the Metropolis-CVAE and the CVAE are identical). CVAE posterior sampling requires (i) a single forward pass through the encoder $E_1$, and (ii) one forward pass per sample through the decoder $D$. This process takes approximately 10 s - 1 min, depending on the number of samples, for the entire data set. This comparison will be added to the paper.
>
> **Reviewer 2.6:** What is the difference between $q$ in Eq.(1) and $\hat{p}$ in Eq.(2)? [...]
>
> **Response 2.6:** $\hat{p}(\mathbf{y}|\mathbf{x})$ is an approximate posterior, while $q(\mathbf{z}|\mathbf{x},\mathbf{y})$ is a so-called recognition function. This is required as part of the factorization of the posterior $p(\mathbf{y}|\mathbf{x})$ in order to transform the problem of maximizing $\log p(\mathbf{y}|\mathbf{x})$ directly - which is intractable - into a tractable inference problem using the stochastic gradient variational Bayes (SGVB) framework. Intuitively, the two terms in Equation (1) can be interpreted as: (i) minimize the Kullback-Leibler divergence between latent space representations computed using full information ($q(\mathbf{z}|\mathbf{x},\mathbf{y})$ has access to both data and labels) and partial information ($p(\mathbf{z}|\mathbf{x})$ has access only to the data, as is the case at inference time), and (ii) maximize the likelihood of the labels given the data, conditional on the latent representation.
>
> **Reviewer 2.7:** In Eq.(10), $L_z$ is not defined.
>
> **Response 2.7:** Thank you, this will be addressed. $L_z$ is the number of samples in the stochastic approximation to the integral in Equation (9).

---

### Official Review · Reviewer_QXph · 2021-07-17

**Rating:** 4
**Confidence:** 4

**Summary:**

This paper introduces an online MCMC framework for semi-supervised learning with variational autoencoders. The idea of the paper stems from physics problems, where one typically wants to infer certain parameters of a data generating process from empirical data. An example of this is quantitative MRI.

The authors consider the model of data generating process known, and also assume a distribution mismatch between the simulated data, as well as the empirical data, which may include certain confounding factors that may be impossible to model. During the the training of a VAE, the authors progressively improve the psudolabels via a Metropolis-Hastings scheme. The key novelty of the paper is that the proposal distribution for the sampler is an posterior distribution $\hat p(\mathbf{y} | \mathbf{x})$, modeled by the VAE itself.


**Limitations And Societal Impact:**

This work does not have any negative societal impact.

My main points for improvements are summarized in the main review. Here is the summary of limitations:

- Lack of SSL baselines
- Lack of clinically-relevant evaluation
- Generalization of the sampler is not investigated (from scan 1 to scan 2)
- Narrow domain in evaluation (the method seems to be general though)
- Lack of insights on distribution shift level
- Only one application: one can also conduct experiments on image classification, and inference of qMRI parameters from knee images.


**Main Review:**

The most interesting part of this paper is a novel application of an existing machine learning result -- its adaptation to the MR physics problem. The release of the code also makes this work rather strong. The manuscript is well written. However, I believe that the experiments in the paper lack insights, rigor, and comparison to alternative methods.

Things to fix:

Equation (7) : please define what is $n$.

Line 177: use \eqref to refer to equation (14)


Experimental design issues:

-- Lack of baselines and proper comparisons. I believe that in order to truly prove the benefits of the proposed technique, one has to collect a sufficiently large dataset, and show that e.g. proposed methods for inference of quantitative MRI parameters boosts the metric of e.g. prediction of brain conditions. I would argue that acceptance rate of the sampler is not the most appropriate metric and substantially limits the level of experimental validation.

-- In the current context, the authors have only MCMC baseline, but what about machine learning methods? E.g. one off-the-shelf semi-supervised baseline could be to train a simple regressor of parameters in a semi-supervised way using e.g. Pi-model or other techniques.

-- How is the method generalizing from being trained on the first scan to the second scan?
- One very critical issue that I see with in paper, is the convergence of the sampler under different domain shift levels. One could test this in a simulator, and then analyze when the method actually stops converging. E.g. visualizing Wassestein distance vs. the level of distribution mismatch would definitely add important insights

- The penultimate issue I have with the experiments in the paper, is that the authors validated only one application: brain. qMRI is investigated in the knee imaging domain. I would propose the authors to strengthen this part as well.

-- The final issue is that the method seems to work for any likelihood function. I would suggest the authors to evaluate it also on CIFAR10/100 classification benchmarks, and compare to classical VAE. The authors can conduct the distribution mismatch experiments there as well.


**Time Spent Reviewing:**

4

---

> ### Author Response · Authors · 2021-08-11
> **Initial response to Reviewer 1 (QXph)**
>
> **Reviewer 1.1:** Equation (7) : please define what is $n$.
>
> **Response 1.1:** $n$ is the most recent sample in the Markov chain of pseudo labels $\tilde{\mathbf{y}}$; this will be clarified in the manuscript.
>
> **Reviewer 1.2:** Line 177: use \eqref to refer to equation (14)
>
> **Response 1.2:** We agree; this will be corrected.
>
> **Reviewer 1.3:** Experimental design issues: [...]
>
> **Response 1.3:** We agree with these experimental design issues, and indeed this was a general sentiment among reviewers; please see our comment addressed to all reviewers.
>
> **Reviewer 1.4:** How is the method generalizing [...]
>
> **Response 1.4:** We have observed that the method generalizes, when trained on only one scan, to the other scan. This is not demonstrated in the paper, but can easily be included in subsequent revisions. Performance degrades compared to when both scans are included, as expected, but the Metropolis-CVAE still outperforms the CVAE.
>
> **Reviewer 1.5:** One very critical issue that I see with in paper, is the convergence of the sampler [...]
>
> **Response 1.5:** We agree that this would generate important insights; such an experiment will be included in future revisions. Please see our comment addressed to all reviewers.
>
> **Reviewer 1.6:** The penultimate issue I have with the experiments in the paper [...]
>
> **Response 1.6:** We agree that more qMRI applications would strengthen the paper. However, we believe that it would be more economical to instead add an application outside of qMRI. That is: keep the current qMRI application, with the addition of a clinically challenging test data (for example pathological data with multiple sclerosis (MS) lesions), add another experiment outside of qMRI, and possibly investigate e.g. knee qMRI as supplementary material. Note however that multi-exponential decay is typically only measured in the brain, and therefore applying our method to knee data would be a strictly simpler problem, with Equation (14) reduced to a single term.
>
> **Reviewer 1.7:** The final issue is that the method seems to work for any likelihood function. [...]
>
> **Response 1.7:** We did consider investigating applications in image classification, and we agree that it would be a useful evaluation of our method. However, it is not clear that a likelihood function is available in this case. For example, one cannot easily answer the question: “Given a class label $\mathbf{y}$, how likely is it that $\mathbf{y}$ corresponds to image $\mathbf{x}$?” - particularly when image $\mathbf{x}$ is unlabeled. Given such an oracle - perhaps arising from a pretrained classifier - it would be possible to use a Metropolis-CVAE to learn to reproduce the distribution of predictions of the oracle, in analogy with student-teacher learning. However, we concluded that this added complexity would somewhat defeat the purpose of the Metropolis-CVAE, which intends to bootstrap labels onto data with well-understood likelihood functions. What perhaps is more within the scope of this work is to investigate inference problems over images (for example, learning to recover an image $\mathbf{y}$ from an image $\mathbf{x}$ corrupted by blur, noise, random crops, etc.). Nevertheless, constructing useful and tractable likelihood functions between images is highly nontrivial; see for example PixelCNN (https://arxiv.org/abs/1606.05328) and PixelCNN++ (https://arxiv.org/pdf/1701.05517.pdf).

---

### Author Response · Authors · 2021-08-11
**General comment to all reviewers**

We would like to thank the reviewers for their insightful and constructive criticisms; we have been provided with many interesting avenues to further explore. In assessing the reviews in aggregate, we note that the fundamental contribution of this work - the Metropolis-CVAE method - was received as a novel and interesting approach. However, there is a consensus among all four reviewers that this work lacks comparisons with semi-supervised learning (SSL) baseline methods, such as the cited method by Kingma et al., as well as lacking sufficient experiments for validation of the method.

It was our original intention to focus this work on the methodology and motivation behind Metropolis-CVAEs, rather than on applications and comparisons with existing SSL methods. Given the consensus of the reviewer responses, we acknowledge that more comparisons with existing SSL methods and experiments are needed. Given the opportunity for subsequent revisions, we propose to incorporate the following comparisons and experiments:
* In addition to the MCMC and vanilla CVAE baselines, the aforementioned SSL method by Kingma et al. described in lines 69-76 of the manuscript will be implemented for all experiments.
* An additional experiment will be added: we proposed to investigate a simple Bayesian linear regression problem subject to distribution shift in the noise model between labeled and unlabeled data. In particular:
    * Let $Y = a X + b + \epsilon$ be a Bayesian linear regression problem with data $\mathbf{x} = (X,Y)$ subject to noise $\epsilon \sim t(\nu)$, where $t(\nu)$ is the Student's $t$-distribution with $\nu$ degrees of freedom
    * Distribution shift will be investigated by:
        1. Generating labeled data $\tilde{p}_{\ell}(\mathbf{x})$ using fixed $\nu = \infty$, i.e. the observations $Y$ are subject to uniform Gaussian noise,
        2. Generating unlabeled data $\tilde{p}^{\nu}_{u}(\mathbf{x})$ parameterized by the number of degrees of freedom $\nu \in [1,\infty)$ of the noise distribution, and
        3. Comparing the performance of each inference method on the unlabelled data sets as a function of $\nu$. In all cases, the likelihood function used will assume a Gaussian noise model, biasing inference when $\nu < \infty$
    * The hypothesis for this experiment is that the Metropolis-CVAE and MCMC methods can reduce inference bias by avoiding the incorrect assumption that $\tilde{p}^{\nu}_{u}(\mathbf{x})$ is equal to the labeled data distribution
* Similar to the above experiment, the first MRI experiment in the current manuscript will be modified to introduce a distribution shift in the unlabeled simulated data:
    * Currently, the MRI parameter $T_1$ in Equation (14) is fixed to $1.0 s$ (line 149).
    * The generation of the unlabeled simulated data set $\tilde{p}_{u,3}(\mathbf{x})$ (lines 177-181) will be modified to incorporate $T_1$ which varies over a prior distribution, while retaining the assumption $T_1 = 1.0s$ in the likelihood function.
    * This modification will simulate distribution shift arising due to an unmodeled physical process.

---

### Decision · Program_Chairs · 2021-09-27

**Decision:**

Reject

**Comment:**

This paper describes an algorithm reminiscent of the dynamics of an Monte-Carlo expectation-maximization (MCEM) algorithm where the E step is replaced by MCMC, here the classical MH algorithm. During the training of a VAE (similar to M step), the authors progressively improve the psudolabels via a Metropolis-Hastings scheme. (The derivation here seems to be pretty similar to and it might be informative if the authors would point out key differences of their approach.) One novelty of the paper is that the proposal distribution for the sampler is a posterior distribution modeled by the VAE itself.

Overall, the reviewers seem to have liked the key idea but raise major concerns regarding the lack of sufficient experimental evaluation at this stage. I think this is a major concern and I encourage the authors to carefully address the reviewers extensive comments and submit to a future venue.